# Phased Exploration with Greedy Exploitation in Stochastic Combinatorial Partial Monitoring Games

**Sougata Chaudhuri**
Department of Statistics
University of Michigan Ann Arbor
`sougata@umich.edu`

**Ambuj Tewari**
Department of Statistics and Department of EECS
University of Michigan Ann Arbor
`tewaria@umich.edu`

## Abstract

Partial monitoring games are repeated games where the learner receives feedback that might be different from adversary's move or even the reward gained by the learner. Recently, a general model of combinatorial partial monitoring (CPM) games was proposed [1], where the learner's action space can be exponentially large and adversary samples its moves from a bounded, continuous space, according to a fixed distribution. The paper gave a confidence bound based algorithm (GCB) that achieves $O(T^{2/3} \log T)$ distribution independent and $O(\log T)$ distribution dependent regret bounds. The implementation of their algorithm depends on two separate offline oracles and the distribution dependent regret additionally requires existence of a unique optimal action for the learner. Adopting their CPM model, our first contribution is a Phased Exploration with Greedy Exploitation (PEGE) algorithmic framework for the problem. Different algorithms within the framework achieve $O(T^{2/3} \sqrt{\log T})$ distribution independent and $O(\log^2 T)$ distribution dependent regret respectively. Crucially, our framework needs only the simpler "argmax" oracle from GCB and the distribution dependent regret does not require existence of a unique optimal action. Our second contribution is another algorithm, PEGE2, which combines gap estimation with a PEGE algorithm, to achieve an $O(\log T)$ regret bound, matching the GCB guarantee but removing the dependence on size of the learner's action space. However, like GCB, PEGE2 requires access to both offline oracles and the existence of a unique optimal action. Finally, we discuss how our algorithm can be efficiently applied to a CPM problem of practical interest: namely, online ranking with feedback at the top.

## 1 Introduction

Partial monitoring (PM) games are repeated games played between a learner and an adversary over discrete time points. At every time point, the learner and adversary each simultaneously select an action, from their respective action sets, and the learner gains a reward, which is a function of the two actions. In PM games, the learner receives limited feedback, which might neither be adversary's move (full information games) nor the reward gained (bandit games). In *stochastic* PM games, adversary generates actions which are independent and identically distributed according to a distribution fixed before the start of the game and unknown to the learner. The learner's objective is to develop a learning strategy that incurs low regret over time, based on the feedback received during the course of the game. Regret is defined as the difference between cumulative reward of the learner's strategy and the best fixed learner's action in hindsight. The usual learning strategies in online games combine some form of exploration (getting feedback on certain learner's actions) and exploitation (playing the perceived optimal action based on current estimates).

Starting with early work in the 2000s [2, 3], the study of *finite* PM games reached a culmination point with a comprehensive and complete classification [4]. We refer the reader to these works for more references and also note that newer results continue to appear [5]. Finite PM games restrict both the learner's and adversary's action spaces to be finite, with a very general feedback model. All finite partial monitoring games can be classified into one of four categories, with minimax regret $\Theta(T)$, $\Theta(T^{2/3})$, $\Theta(T^{1/2})$ and $\Theta(1)$. The classification is governed by *global* and *local observability* properties pertaining to a game [4]. Another line of work has extended traditional multi-armed bandit problem (MAB) [6] to include combinatorial action spaces for learner (CMAB) [7, 8]. The combinatorial action space can be exponentially large, rendering traditional MAB algorithms designed for small finite action spaces, impractical with regret bounds scaling with size of action space. The CMAB algorithms exploit a finite subset of base actions, which are specific to the structure of problem at hand, leading to practical algorithms and regret bounds that do not scale with, or scale very mildly with, the size of the learner's action space.

While finite PM and CMAB problems have witnessed a lot of activity, there is only one paper [1] on combinatorial partial monitoring (CPM) games, to the best of our knowledge. In that paper, the authors combined the combinatorial aspect of CMAB with the limited feedback aspect of finite PM games, to develop a CPM model. The model extended PM games to include combinatorial action spaces for learner, which might be exponentially large, and infinite action spaces for the adversary. Neither of these situations can be handled by generic algorithms for finite PM games. Specifically, the model considered an action space $\mathcal{X}$ for the learner, that has a small subset of actions defining a *global observable set* (see Assumption 2 in Section 2). The adversary's action space is a continuous, bounded vector space with the adversary sampling moves from a fixed distribution over the vector space. The reward function considered is a general non-linear function of learner's and adversary's actions, with some restrictions (see Assumptions 1 & 3 in Section 2). The model incorporated a linear feedback mechanism where the feedback received is a linear transformation of adversary's move. Inspired by the classic confidence bound algorithms for MABs, such as UCB [6], the authors proposed a Global Confidence Bound (GCB) algorithm that enjoyed two types of regret bound. The first one was a distribution independent $O(T^{2/3} \log T)$ regret bound and the second one was a distribution dependent $O(\log T)$ regret bound. A distribution dependent regret bound involves factors specific to the adversary's fixed distribution, while distribution independent means the regret bound holds over all possible distributions in a broad class of distributions. Both bounds also had a logarithmic dependence on $|\mathcal{X}|$. The algorithm combined online estimation with two offline computational oracles. The first oracle finds the action(s) achieving maximum value of reward function over $\mathcal{X}$, for a particular adversary action (argmax oracle), and the second oracle finds the action(s) achieving second maximum value of reward function over $\mathcal{X}$, for a particular adversary action (arg-secondmax oracle). Moreover, the distribution dependent regret bound requires existence of a *unique* optimal learner action. The inspiration for the CPM model came from various applications like crowdsourcing and matching problems like matching products with customers.

**Our Contributions.** We adopt the CPM model proposed earlier [1]. However, instead of using upper confidence bound techniques, our work is motivated by another classic technique developed for MABs, namely that of forced exploration. This technique was already used in the classic paper of Robbins [9] and has also been called "forcing with certainty equivalence" in the control theory literature [10]. We develop a Phased Exploration with Greedy Exploitation (PEGE) algorithmic framework (Section 3) borrowing the PEGE terminology from work on linearly parameterized bandits [11]. When the framework is instantiated with different parameters, it achieves $O(T^{2/3}\sqrt{\log T})$ distribution independent and $O(\log^2 T)$ distribution dependent regret. Significantly, the framework combines online estimation with only the argmax oracle from GCB, which is a practical advantage over requiring an additional arg-secondmax oracle. Moreover, the distribution dependent regret does not require existence of unique optimal action. Uniqueness of optimal action can be an unreasonable assumption, especially in the presence of a combinatorial action space. Our second contribution is another algorithm PEGE2 (Section 4) that combines a PEGE algorithm with Gap estimation, to achieve a distribution dependent $O(\log T)$ regret bound, thus matching the GCB regret guarantee in terms of $T$ and gap. Here, gap refers to the difference between expected reward of optimal and second optimal learner's actions. However, like GCB, PEGE2 does require access to both the oracles, existence of unique optimal action for $O(\log T)$ regret and its regret is never larger than $O(T^{2/3} \log T)$ when there is no unique optimal action. A crucial advantage of PEGE and PEGE2 over GCB is that all our regret bounds are independent of $|\mathcal{X}|$, only depending on the size of the

small *global observable set*. Thus, though we have adopted the CPM model [1], our regret bounds are meaningful for countably infinite or even continuous learner's action space, whereas GCB regret bound has an explicit logarithmic dependence on $|\mathcal{X}|$. We provide a detailed comparison of our work with the GCB algorithm in Section 5. Finally, we discuss how our algorithms can be efficiently applied in the CPM problem of online ranking with feedback restricted to top ranked items (Section 6), a problem already considered [12] but analyzed in a non-stochastic setting.

## 2  Preliminaries and Assumptions

The online game is played between a learner and an adversary, over discrete rounds indexed by $t = 1, 2, \ldots$. The learner's action set is denoted as $\mathcal{X}$ which can be exponentially large. The adversary's action set is the infinite set $[0, 1]^n$. The adversary fixes a distribution $p$ on $[0, 1]^n$ before start of the game (adversary's strategy), with $p$ unknown to the learner. At each round of the game, adversary samples $\theta(t) \in [0, 1]^n$ according to $p$, with $\mathbb{E}_{\theta(t) \sim p}[\theta(t)] = \theta_p^*$. The learner chooses $x(t) \in \mathcal{X}$ and gets reward $r(x(t), \theta(t))$. However, the learner might not get to know either $\theta(t)$ (as in a full information game) or $r(x(t), \theta(t))$ (as in a bandit game). In fact, the learner receives, as feedback, a linear transformation of $\theta(t)$. That is, every action $x \in \mathcal{X}$ has an associated transformation matrix $M_x \in \mathbb{R}^{m_x \times n}$. On playing action $x(t)$, the learner receives a feedback $M_{x(t)} \cdot \theta(t) \in \mathbb{R}^{m_x}$. Note that the game with the defined feedback mechanism subsumes full information and bandit games. $M_x = \mathbb{I}^{n \times n}, \ \forall x$ makes it a full information game since $M_x \cdot \theta = \theta$. If $r(x, \theta) = x \cdot \theta$, then $M_x = x \in \mathbb{R}^n$ makes it a bandit game. The dimension $n$, action space $\mathcal{X}$, reward function $r(\cdot, \cdot)$ and transformation matrices $M_x, \forall x \in \mathcal{X}$ are known to the learner. The goal of the learner is to minimize the expected regret, which, for a given time horizon $T$, is:

$$R(T) = T \cdot \max_{x \in \mathcal{X}} \mathbb{E}[r(x, \theta)] - \sum_{t=1}^{T} \mathbb{E}[r(x(t), \theta(t))] \tag{1}$$

where the expectation in the first term is taken over $\theta$, w.r.t. distribution $p$, and the second expectation is taken over $\theta$ and possible randomness in the learner's algorithm.

**Assumption 1. (Restriction on Reward Function)** The first assumption is that $\mathbb{E}_{\theta \sim p}[r(x, \theta)] = \bar{r}(x, \theta_p^*)$, for some function $\bar{r}(\cdot, \cdot)$. That is, the expected reward is a function of $x$ and $\theta_p^*$, which is always satisfied if $r(x, \theta)$ is a linear function of $\theta$, or if distribution $p$ happens to be any distribution with support $[0, 1]^n$ and fully parameterized by its mean $\theta_p^*$. With this assumption, the expected regret becomes:

$$R(T) = T \cdot \bar{r}(x^*, \theta_p^*) - \sum_{t=1}^{T} \mathbb{E}[\bar{r}(x(t), \theta_p^*)]. \tag{2}$$

For distribution dependent regret bounds, we define gaps in expected rewards: Let $x^* \in S(\theta_p^*) = \text{argmax}_{x \in X} \bar{r}(x, \theta_p^*)$. Then $\Delta_x = \bar{r}(x^*, \theta_p^*) - \bar{r}(x, \theta_p^*)$, $\Delta_{max} = \max\{\Delta_x : x \in \mathcal{X}\}$ and $\Delta = \min\{\Delta_x : x \in \mathcal{X}, \Delta_x > 0\}$.

**Assumption 2. (Existence of Global Observable Set)** The second assumption is on the existence of a *global observable set*, which is a subset of learner's action set and is required for estimating an adversary's move $\theta$. The *global observable set* is defined as follows: for a set of actions $\sigma = \{x_1, x_2, \ldots, x_{|\sigma|}\} \subseteq \mathcal{X}$, let their transformation matrices be stacked in a top down fashion to obtain a $\mathbb{R}^{\sum_{i=1}^{|\sigma|} m_{x_i} \times n}$ dimensional matrix $M_\sigma$. $\sigma$ is said to be a global observable set if $M_\sigma$ has full column rank, i.e., $\text{rank}(M_\sigma) = n$. Then, the Moore-Penrose pseudoinverse $M_\sigma^+$ satisfies $M_\sigma^+ M_\sigma = \mathbb{I}^{n \times n}$. Without the assumption on the existence of global observable set, it might be the case that even if the learner plays all actions in $\mathcal{X}$ on same $\theta$, the learner might not be able to recover $\theta$ (as $M_\sigma^+ M_\sigma = \mathcal{I}^{n \times n}$ will not hold without full rank assumption). In that case, learner might not be able to distinguish between $\theta_{p_1}^*$ and $\theta_{p_2}^*$, corresponding to two different adversary's strategies. Then, with non-zero probability, the learner can suffer $\Omega(T)$ regret and no learner strategy can guarantee a sub-linear in $T$ regret (the intuition forms the base of the *global observability condition* in [2]). Note that the size of the global observable set is small, i.e., $|\sigma| \leq n$. A global observable set can be found by including an action $x$ in $\sigma$ if it strictly increases the rank of $M_\sigma$, till the rank reaches $n$. There can, of course, be more than one global observable set.

**Assumption 3. (Lipschitz Continuity of Expected Reward Function)** The third assumption is on the Lipschitz continuity of expected reward function in its second argument. More precisely, it is assumed that $\exists\, R > 0$ such that $\forall\, x \in \mathcal{X}$, for any $\theta_1$ and $\theta_2$, $|\bar{r}(x, \theta_1) - \bar{r}(x, \theta_2)| \leq R\|\theta_1 - \theta_2\|_2$. This assumption is reasonable since otherwise, a small error in estimation of mean reward vector $\theta_p^*$ can introduce a large change in expected reward, leading to difficulty in controlling regret over time. The Lipschitz condition holds trivially for expected reward functions which are linear in second argument. The continuity assumption, along with the fact that adversary's moves are in $[0, 1]^n$, implies boundedness of expected reward for any learner's action and any adversary's action. We denote $R_{max} = \max_{x \in \mathcal{X}, \theta \in [0,1]^n} \bar{r}(x, \theta)$.

The three assumptions above will be made throughout. However, the fourth assumption will only be made in a subset of our results.

**Assumption 4. (Unique Optimal Action)** The optimal action $x^* = \operatorname{argmax}_{x \in \mathcal{X}} \bar{r}(x, \theta_p^*)$ is unique. Denote a second best action (which may not be unique) by $x_-^* = \operatorname{argmax}_{x \in \mathcal{X}, x \neq x^*} \bar{r}(x, \theta_p^*)$. Note that $\Delta = \bar{r}(x^*, \theta_p^*) - \bar{r}(x_-^*, \theta_p^*)$.

## 3 Phased Exploration with Greedy Exploitation

Algorithm 1 (PEGE) uses the classic idea of doing exploration in phases that are successively further apart from each other. In between exploration phases, we select action greedily by completely trusting the current estimates. The constant $\beta$ controls how much we explore in a given phase and the constant $\alpha$ along with the function $C(\cdot)$ determines how much we exploit. This idea is classic in the bandit literature [9–11] but has not been applied to the CPM framework to the best of our knowledge.

---

**Algorithm 1** The PEGE Algorithmic Framework

---

1: Inputs: $\alpha$, $\beta$ and function $C(\cdot)$ (to determine amount of exploration/exploitation in each phase).

2: For $b = 1, 2, \ldots,$
3:     **Exploration**
4:         For $i = 1$ to $|\sigma|$ ($\sigma$ is global observable set)
5:             For $j = 1$ to $b^\beta$
6:                 Let $t_{j,i} = t$ and $\theta(t_{j,i}, b) = \theta(t)$ where $t$ is current time point
7:                 Play $x_i \in \sigma$ and get feedback $M_{x_i} \cdot \theta(t_{j,i}, b) \in \mathbb{R}^{m_{x_i}}$.
8:             End For
9:         End For
10:    **Estimation**
11:     $\tilde{\theta}_{j,i} = M_\sigma^+(M_{x_1} \cdot \theta(t_{j,1}, i), \ldots, M_{x_{|\sigma|}} \cdot \theta(t_{j,|\sigma|}, i)) \in \mathbb{R}^n$.
12:     $\hat{\theta}(b) = \dfrac{\sum_{i=1}^{b} \sum_{j=1}^{i^\beta} \tilde{\theta}_{j,i}}{\sum_{j=1}^{b} j^\beta} \in \mathbb{R}^n$.
13:     $x(b) \in \operatorname{argmax}_{x \in \mathcal{X}} \bar{r}(x, \hat{\theta}(b))$.
14:    **Exploitation**
15:         For $i = 1$ to $\exp(C(b^\alpha))$
16:             Play $x(b)$.
17:         End For
18: End For

---

It is easy to see that the estimators in Algorithm 1 have the following properties: $\mathbb{E}_p[\tilde{\theta}_{j,i}] = M_\sigma^+(M_{x_1} \cdot \theta_p^*, \ldots, M_{x_{|\sigma|}} \cdot \theta_p^*) = M_\sigma^+ M_\sigma \cdot \theta_p^* = \theta_p^*$ and hence $\mathbb{E}_p[\hat{\theta}] = \theta_p^*$. Using the fact that $M_\sigma^+ = (M_\sigma^\top M_\sigma)^{-1} M_\sigma^\top$, we also have the following bound on estimation error of $\theta_p^*$:

$$\|\tilde{\theta}_{j,i} - \theta_p^*\|_2 \leq \|M_\sigma^+(M_{x_1} \cdot \theta(t_{j,1}, i), \ldots, M_{x_{|\sigma|}} \cdot \theta(t_{j,|\sigma|}, i)) - M_\sigma^+ M_\sigma \theta_p^*\|_2$$

$$= \|(M_\sigma^\top M_\sigma)^{-1} \sum_{k=1}^{|\sigma|} M_{x_k}^\top M_{x_k} \cdot (\theta(t_{j,k}, i) - \theta_p^*)\|_2 \leq \sqrt{n} \sum_{k=1}^{|\sigma|} \|(M_\sigma^\top M_\sigma)^{-1} M_{x_k}^\top M_{x_k}\|_2 =: \beta_\sigma$$

$$\tag{3}$$

where the constant $\beta_\sigma$ defined above depends only on the structure of the linear transformation matrices of the global observer set and not on adversary strategy $p$.

Our first result is about the regret of Algorithm 1 when within phase number $b$, the exploration part spends $|\sigma|$ rounds (constant w.r.t. $b$) and the exploitation part grows polynomially with $b$.

**Theorem 1. (Distribution Independent Regret)** *When Algorithm 1 is initialized with the parameters $C(a) = \log a$, $\alpha = 1/2$ and $\beta = 0$, and the online game is played over $T$ rounds, we get the following bound on expected regret:*

$$R(T) \le R_{max}|\sigma|T^{2/3} + 2R\beta_\sigma T^{2/3}\sqrt{\log 2e^2 + 2\log T} + R_{max} \tag{4}$$

*where $\beta_\sigma$ is the constant as defined in Eq. 3.*

Our next result is about the regret of Algorithm 1 when within phase number $b$, the exploration part spends $|\sigma| \cdot b$ rounds (linearly increasing with $b$) and the exploitation part grows exponentially with $b$.

**Theorem 2. (Distribution Dependent Regret)** *When Algorithm 1 is initialized with the parameters $C(a) = h \cdot a$, for a tuning parameter $h > 0$, $\alpha = 1$ and $\beta = 1$, and the online game is played over $T$ rounds, we get the following bound on expected regret:*

$$R(T) \le \sum_{x \in \sigma} \Delta_x \left(\frac{\log T}{h}\right)^2 + \frac{4\sqrt{2\pi}e^2 R\Delta_{max}\beta_\sigma}{\Delta} e^{\frac{h^2(2R^2\beta_\sigma^2)}{\Delta^2}}. \tag{5}$$

Such an explicit bound for a PEGE algorithm that is polylogarithmic in $T$ and explicitly states the multiplicative and additive constants involved in not known, to the best of our knowledge, even in the bandit literature (e.g., earlier bounds [10] are asymptotic) whereas here we prove it in the CPM setting. Note that the additive constant above, though finite, blows up exponentially fast as $\Delta \to 0$ for a fixed $h$. It is well behaved however, if the tuning parameter $h$ is on the same scale as $\Delta$. This line of thought motivates us to estimate the gap to within constant factors and then feed that estimate into a PEGE algorithm. This is what we will do in the next section.

## 4 Combining Gap Estimation with PEGE

Algorithm 2 tries to estimate the gap $\Delta$ to within a constant multiplicative factor. However, if there is no unique optimal action or when the true gap is small, gap estimation can take a very large amount of time. To prevent that from happening, the algorithm also takes in a threshold $T_0$ as input and definitely stops if the threshold is reached. The result below assures us that, with high probability, the algorithm behaves as expected. That is, if there is a unique optimal action and the gap is large enough to be estimated with a given confidence before the threshold $T_0$ kicks in, it will output an estimate $\hat{\Delta}$ in the range $[0.5\Delta, 1.5\Delta]$. On the other hand, if there is no unique optimal action, it does not generate an estimate of $\Delta$ and instead runs out of the exploration budget $T_0$.

**Theorem 3. (Gap Estimation within Constant Factors)** *Let $T_0 \ge 1$ and $\delta \in (0, 1)$ and define $T_1(\delta) = \frac{256R^2\beta_\sigma^2}{\Delta^2}\log\frac{512e^2R^2\beta_\sigma^2}{\Delta^2\delta}$, $T_2(\delta) = \frac{16R^2\beta_\sigma^2}{\Delta^2}\log\frac{4e^2}{\delta}$. Consider Algorithm 2 run with*

$$w(b) = \sqrt{\frac{R^2\beta_\sigma^2\log(\frac{4e^2b^2}{\delta})}{b}}. \tag{6}$$

*Then, the following 3 claims hold.*

1. *Suppose Assumption 4 holds and $T_1(\delta) < T_0$. Then with probability at least $1 - \delta$, Algorithm 2 stops in $T_1(\delta)$ episodes and outputs an estimate $\hat{\Delta}$ that satisfies $\frac{1}{2}\Delta \le \hat{\Delta} \le \frac{3}{2}\Delta$.*

2. *Suppose Assumption 4 holds and $T_0 \le T_1(\delta)$. Then with probability at least $1 - \delta$, the algorithm either outputs "threshold exceeded" or outputs an estimate $\hat{\Delta}$ that satisfies $\frac{1}{2}\Delta \le \hat{\Delta} \le \frac{3}{2}\Delta$. Furthermore, if it outputs $\hat{\Delta}$, it must be the case that the algorithm stopped at an episode $b$ such that $T_2(\delta) < b < T_0$.*

3. *Suppose Assumption 4 fails. Then, with probability at least $1 - \delta$, Algorithm 2 stops in $T_0$ episodes and outputs "threshold exceeded".*

---
**Algorithm 2** Algorithm for Gap Estimation
---
1: Inputs: $T_0$ (exploration threshold) and $\delta$ (confidence parameter)

2: For $b = 1, 2, \ldots,$
3:     **Exploration**
4:         For $i = 1$ to $|\sigma|$
5:             (Denote) $t_i = t$ and $\theta(t_i, b) = \theta(t)$ ($t$ is current time point).
6:             Play $x_i \in \sigma$ and get feedback $M_{x_i} \cdot \theta(t_i, b) \in \mathbb{R}^{m_{x_i}}$.
7:         End For
8:     **Estimation**
9:        $\tilde{\theta}_b = M_\sigma^+(M_{x_1} \cdot \theta(t_1, b), \ldots, M_{x_{|\sigma|}} \cdot \theta(t_{|\sigma|}, b)) \in \mathbb{R}^n$.
10:     $\hat{\theta}(b) = \dfrac{\sum_{i=1}^{b} \tilde{\theta}_i}{b} \in \mathbb{R}^n$.

11:     **Stopping Rule** ($w(b)$ is defined as in Eq. (6))
12:     If $\operatorname{argmax}_{x \in \mathcal{X}} \bar{r}(x, \hat{\theta}(b))$ is unique:
13:         $\hat{x}(b) = \operatorname{argmax}_{x \in \mathcal{X}} \bar{r}(x, \hat{\theta}(b))$
14:         $\hat{x}_-(b) = \operatorname{argmax}_{x \in \mathcal{X}, x \neq \hat{x}(b)} \bar{r}(x, \hat{\theta}(b))$ (need not be unique)
15:         If $\bar{r}(\hat{x}(b), \hat{\theta}(b)) - \bar{r}(\hat{x}_-(b), \hat{\theta}(b)) > 6w(b)$:
16:             STOP and output $\hat{\Delta} = \bar{r}(\hat{x}(b), \hat{\theta}(b)) - \bar{r}(\hat{x}_-(b), \hat{\theta}(b))$
17:         End If
18:     End If
19:     If $b > T_0$:
20:         STOP and output "threshold exceeded"
21:     End If
22: End For

---

Equipped with Theorem 3, we are now ready to combine Algorithm 2 with Algorithm 1 to give Algorithm 3. Algorithm 3 first calls Algorithm 2. If Algorithm 2 outputs an estimate $\hat{\Delta}$ it is fed into Algorithm 1. If the threshold $T_0$ is exceeded, then the remaining time is spent in pure exploitation. Note that by choosing $T_0$ to be of order $T^{2/3}$ we can guarantee a worst case regret of the same order even when unique optimality assumption fails. For PM games that are globally observable but not locally observable, such a distribution independent $O(T^{2/3})$ bound is known to be optimal [4].

**Theorem 4. (Regret Bound for PEGE2)** *Consider Algorithm 3 run with knowledge of the number $T$ of rounds. Consider the distribution independent bound*

$$B_1(T) = 2(2R\beta_\sigma |\sigma|^2 R_{max}^2 T)^{2/3} \sqrt{\log(4e^2 T^3)} + R_{max},$$

*and the distribution dependent bound*

$$B_2(T) = \frac{256 R^2 \beta_\sigma^2}{\Delta^2} \log \frac{512 e^2 R^2 \beta_\sigma^2 T}{\Delta^2} R_{max} |\sigma| + \sum_{x \in \sigma} \Delta_x \frac{36 R^2 \beta_\sigma^2 \log T}{\Delta^2} + \frac{8e^2 R^2 \beta_\sigma^2}{\Delta^2} + R_{max}.$$

*If Assumption 4 fails, then the expected regret of Algorithm 3 is bounded as $R(T) \leq B_1(T)$. If Assumption 4 holds, then the expected regret of Algorithm 3 is bounded as*

$$R(T) \leq \begin{cases} B_2(T) & \text{if } T_1(\delta) < T_0 \\ O(T^{2/3} \log T) & \text{if } T_0 \leq T_1(\delta) \end{cases}, \tag{7}$$

*where $T_1(\delta)$ is as defined in Theorem 3 and $\delta, T_0$ are as defined in Algorithm 3.*

In the above theorem, note that $T_1(\delta)$ scales as $\Theta(\frac{1}{\Delta^2} \log \frac{T}{\Delta^2})$ and $T_0$ as $\Theta(T^{2/3})$. Thus, the two cases in Eq. (7) correspond to large gap and small gap situations respectively.

## 5   Comparison with GCB Algorithm

We provide a detailed comparison of our results with those obtained for GCB [1]. (a) While we use the same CPM model, our solution is inspired by the forced exploration technique while GCB

**Algorithm 3** Algorithm Combining PEGE with Gap Estimation (PEGE2)

---

1: Input: $T$ (total number of rounds)

2: Call Algorithm 2 with inputs $T_0 = \left( \frac{2R\beta_\sigma T}{|\sigma|R_{max}} \right)^{2/3}$ and $\delta = 1/T$
3: If Algorithm 2 returns "threshold exceeded":
4:     Let $\hat{\theta}(T_0)$ be the latest estimate of $\theta_p^*$ maintained by Algorithm 2
5:     Play $\hat{x}(T_0) = \text{argmax}_{x \in \mathcal{X}} \, \bar{r}(x, \hat{\theta})$ for the remaining $T - T_0|\sigma|$ rounds
6: Else:
7:     Let $\hat{\Delta}$ be the gap estimate produced by Algorithm 2
8:     For all remaining time steps, run Algorithm 1 with parameters $C(a) = ha$ with
    $h = \frac{\hat{\Delta}^2}{9R^2\beta_\sigma^2}, \alpha = 1, \beta = 0$
9: End If

---

is inspired by the confidence bound technique, both of which are classic in the bandit literature. (b) One instantiation of our PEGE framework gives an $O(T^{2/3}\sqrt{\log T})$ distribution independent regret bound (Theorem 1), which does not require call to arg-secondmax oracle. This is of substantial practical advantage over GCB since even for linear optimization problems over polyhedra, standard routines usually do not have option of computing action(s) that achieve second maximum value for the objective function. (c) Another instantiation of the PEGE framework gives an $O(\log^2 T)$ distribution dependent regret bound (Theorem 2), which neither requires call to arg-secondmax oracle nor the assumption of existence of unique optimal action for learner. This is once again important, since the assumption of existence of unique optimal action might be impractical, especially for exponentially large action space. However, the caveat is that improper setting of the tuning parameter $h$ in Theorem 2 can lead to an exponentially large additive component in the regret. (d) A crucial point, which we had highlighted in the beginning, is that the regret bounds achieved by PEGE and PEGE2 do not have dependence on size of learner's action space, i.e., $|\mathcal{X}|$. The dependence is only on the size of global observable set $\sigma$, which is guaranteed to be not more than dimension of adversary's action space. Thus, though we have adopted the CPM model [1], our algorithms achieve meaningful regret bounds for countably infinite or even continuous learner's action space. In contrast, the GCB regret bounds have explicit, logarithmic dependence on size of learner's action space. Thus, their results cannot be extended to problems with infinite learner's action space (see Section 6 for an example), and are restricted to large, but *finite* action spaces. (e) The PEGE2 algorithm is a true analogue of the GCB algorithm, matching the regret bounds of GCB in terms of $T$ and gap $\Delta$ with the advantage that it has no dependence on $|\mathcal{X}|$. The disadvantage, however, is that PEGE2 requires knowledge of time horizon $T$, while GCB is an anytime algorithm. It remains an open problem to design an algorithm that combines the strengths of PEGE2 and GCB.

## 6   Application to Online Ranking

A recent paper studied the problem of online ranking with feedback restricted to top ranked items [12]. The problem was studied in a non-stochastic setting, i.e., it was assumed that an oblivious adversary generates reward vectors. Moreover, the learner's action space was exponentially large in number of items to be ranked. The paper made the connection of the problem setting to PM games (but not combinatorial PM games) and proposed an efficient algorithm for the specific problem at hand. However, a careful reading of the paper shows that their algorithmic techniques can handle the CPM model we have discussed so far, but in the *non-stochastic* setting. The reward function is linear in both learner's and adversary's moves, adversary's move is restricted to a finite space of vectors and feedback is a linear transformation of adversary's move. In this section, we give a brief description of the problem setting and show how our algorithms can be used to efficiently solve the problem of online ranking with feedback on top ranked items in the *stochastic* setting. We also give an example of how the ranking problem setting can be somewhat naturally extended to one which has continuous action space for learner, instead of large but finite action space.

The paper considered an online ranking problem, where a learner repeatedly re-ranks a set of $n$, fixed items, to satisfy diverse users' preferences, who visit the system sequentially. Each learner action $x$

is a permutation of the $n$ items. Each user has like/dislike preference for each item, varying between users, with each user's preferences encoded as an $n$ length binary relevance vector $\theta$. Once the ranked list of items is presented to the user, the user scans through the items, but gives relevance feedback only on top ranked item. However, the performance of the learner is judged based on full ranked list and unrevealed, full relevance vector. Thus, we have a PM game, where neither adversary generated relevance vector nor reward is revealed to learner. The paper showed how a number of practical ranking measures, like Discounted Cumulative Gain (DCG), can be expressed as a linear function, i.e., $r(x, \theta) = f(x) \cdot \theta$. The practical motivation of the work was based on learning a ranking strategy to satisfy diverse user preferences, but with limited feedback received due to user burden constraints and privacy concerns.

**Online Ranking with Feedback at Top as a Stochastic CPM Game.** We show how our algorithms can be applied in online ranking with feedback for top ranked items by showing how it is a specific instance of the CPM model and how our key assumptions are satisfied. The learner's action space is the finite but exponentially large space of $\mathcal{X} = n!$ permutations. Adversary's move is an $n$ dimensional relevance vector, and thus, is restricted to $\{0, 1\}^n$ (finite space of size $2^n$) contained in $[0, 1]^n$. In the stochastic setting, we can assume that adversary samples $\theta \in \{0, 1\}^n$ from a fixed distribution on the space. Since the feedback on playing a permutation is the relevance of top ranked item, each move $x$ has an associated transformation matrix (vector) $M_x \in \{0, 1\}^n$, with 1 in the place of the item which is ranked at the top by $x$ and 0 everywhere else. Thus, $M_x \cdot \theta$ gives the relevance of item ranked at the top by $x$. The global observable set $\sigma$ is the set of any $n$ actions, where each action, in turn, puts a distinct item on top. Hence, $M_\sigma$ is the $n \times n$ dimensional permutation matrix. Assumption 1 is satisfied because the reward function is linear in $\theta$ and $\bar{r}(x, \theta_p^*) = f(x) \cdot \theta_p^*$, where $\mathbb{E}_p[\theta] = \theta_p^* \in [0, 1]^n$. Assumption 2 is satisfied since there will always be a global observable set of size $n$ and can be found easily. In fact, there will be multiple global observable sets, with the freedom to choose any one of them. Assumption 3 is satisfied due to the expected reward function being linear in second argument. The Lipschitz constant is $\max_{x \in \mathcal{X}} \|f(x)\|_2$, which is always less than some small polynomial factor of $n$, depending on specific $f(\cdot)$. The value of $\beta_\sigma$ can be easily seen to be $n^{3/2}$. The argmax oracle returns the permutation which simply sorts items according to their corresponding $\theta$ values. The arg-secondmax oracle is more complicated, though feasible. It requires first sorting the items according to $\theta$ and then compare each pair of consecutive items to see where least drop in reward value occurs and switch the corresponding items.

**Likely Failure of Unique Optimal Action Assumption.** Assumption 4 is unlikely to hold in this problem setting (though of course theoretically possible). The mean relevance vector $\theta_p^*$ effectively reflects the average preference of all users for each of the $n$ items. It is very likely that at least a few items will not be liked by anyone and which will ultimately be always ranked at the bottom. Equally possible is that two items will have same user preference on average, and can be exchanged without hurting the optimal ranking. Thus, existence of an unique optimal ranking, which indicates that each item will have different average user preference than every other item, is unlikely. Thus, PEGE algorithm can still be applied to get poly-logarithmic regret (Theorem 2), but GCB will only achieve $O(T^{2/3} \log T)$ regret.

**A PM Game with Infinite Learner Action Space.** We give a simple modification of the ranking problem above to show how the learner can have continuous action space. The learner now ranks the items by producing an $n$ dimensional score vector $x \in [0, 1]^n$ and sorting items according to their scores. Thus the learner's action space is now an uncountably infinite continuous space. As before, the user gets to see the ranked list and gives relevance feedback on top ranked item. The learner's performance will now be judged by a continuous loss function, instead of a discrete-valued ranking measure, since its moves are in a continuous space. Consider the simplest loss, viz., the squared "loss" $r(x, \theta) = -\|x - \theta\|_2^2$ (note -ve sign to keep reward interpetation). It can be easily seen that $\bar{r}(x, \theta_p^*) = E_{\theta \sim p}[r(x, \theta)] = -\|x\|_2^2 + 2x \cdot \theta_p^* - \mathbf{1} \cdot \theta_p^*$, if the relevance vectors $\theta$ are in $\{0, 1\}^n$. Thus, the Lipschitz condition is satisfied. The global observable set is still of size $n$, with the $n$ actions being any $n$ score vectors, whose sorted orders place each of the $n$ items, in turn, on top. $\beta_\sigma$ remains same as before, with $\operatorname{argmax}_x \mathbb{E}_{\theta \sim p} r(x, \theta) = \mathbb{E}_{\theta \sim p}[\theta] = \theta_p^*$. Both PEGE and PEGE2 can achieve meaningful regret bound for this problem, while GCB cannot.

### Acknowledgements

We acknowledge the support of NSF via grants IIS 1452099 and CCF 1422157.

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
