[Supplementary Material]

# 7 Appendix

We first state the large deviation inequality for vector-valued martingales, which is the generalization of Azuma-Hoeffding inequality for scalar valued martingales.

**Theorem 1.8 of [13]**: Let $X_0, X_1, \ldots, X_m$ be a weak martingale sequence taking values in euclidean space $\mathbb{R}^d$, with $\mathbb{E}[X_i|X_{i-1}] = X_{i-1}$. Let $X_0 = 0$ and $\|X_i - X_{i-1}\|_2 \leq 1$, for $i = 1, 2, \ldots, m$. Then, for every $\epsilon > 0$,

$$\Pr[\|X_m\|_2 \geq \epsilon] < 2e^2 e^{\frac{-\epsilon^2}{m}} \tag{8}$$

We use the concentration inequality to get a uniform confidence bound, over the space of learner's action, on the deviation of estimated reward from true reward, after each estimate of mean reward vector is produced.

**Lemma 5.** *At the end of exploration phase within phase $b$, $b = 1, 2, \ldots$, of Algorithm PEGE, the estimator of reward vector $\theta_p^*$ is $\hat{\theta}(b) = \dfrac{\sum_{i=1}^{b} \sum_{j=1}^{i^\beta} \tilde{\theta}_{j,i}}{\sum_{j=1}^{b} j^\beta}$. Then, $\forall \eta > 0$,*

$$\Pr[\forall x \in \mathcal{X} : |\bar{r}(x, \hat{\theta}(b)) - \bar{r}(x, \theta_p^*)| \leq \eta] \geq 1 - 2e^2 e^{\frac{-(\sum_{i=1}^{b^\beta} i^\beta)\eta^2}{R^2 \beta_\sigma^2}} \tag{9}$$

*where $\beta_\sigma$ is the constant as defined in Eq. 3 and $R$ is the Lipschitz constant defined in Assumption 3.*

*Proof.* Let $\{X_{i,j}\}_{\substack{j=1,\ldots,i^\beta \\ i=1,\ldots,b}}$ be a sequence of random vectors, defined as follows:

$$X_{i,j} = \frac{\sum_{i'=1}^{i-1} \sum_{j'=1}^{i'^\beta} \theta_p^* + \sum_{k=1}^{j} \theta_p^* - (\sum_{i'=1}^{i-1} \sum_{j'=1}^{i'^\beta} \tilde{\theta}_{j',i'} + \sum_{k=1}^{j} \tilde{\theta}_{k,i})}{\sum_{i''=1}^{b} \sum_{j''=1}^{(i'')^\beta} \beta_\sigma} \tag{10}$$

It can be checked that the $\ell_2$ norm of the difference between any two consecutive random vectors is bounded by a constant. That is, $\|X_{i,j} - X_{i,j-1}\|_2 = \dfrac{\|\theta_p^* - \tilde{\theta}_{j,i}\|_2}{\sum_{i''=1}^{b} \sum_{j''=1}^{(i'')^\beta} \beta_\sigma} \leq \dfrac{1}{\sum_{i''=1}^{b}(i'')^\beta}$ and

$\|X_{i+1,1} - X_{i,i^\beta}\|_2 \dfrac{\|\theta_p^* - \tilde{\theta}_{1,i+1}\|_2}{\sum_{i''=1}^{b} \sum_{j''=1}^{(i'')^\beta} \beta_\sigma} \leq \dfrac{1}{\sum_{i''=1}^{b}(i'')^\beta}$.

Also, $\tilde{\theta}_{j,i}$ is independent of all estimators formed before $\tilde{\theta}_{j,i}$ in Algorithm PEGE. Thus,

$$\begin{aligned}
\mathbb{E}[X_{i,j} - X_{i,j-1}|X_{i,j-1}] &= \mathbb{E}\left[\frac{\theta_p^* - \tilde{\theta}_{j,i}}{\sum_{i''=1}^{b} \sum_{j''=1}^{(i'')^\beta} \beta_\sigma}|X_{i,j-1}\right] \\
&= \mathbb{E}\left[\frac{\theta_p^* - \tilde{\theta}_{j,i}}{\sum_{i''=1}^{b} \sum_{j''=1}^{(i'')^\beta} \beta_\sigma}\right] = 0
\end{aligned} \tag{11}$$

Thus, $\{X_{i,j}\}_{\substack{j=1,\ldots,i^\beta \\ i=1,\ldots,b}}$ satisfy the criteria of weak martingale sequence and hence, by the large deviation inequality of vector valued martingales, we have,

$$\forall \epsilon > 0, \ \Pr[\|X_{b,b^\beta}\|_2 \geq \epsilon] < 2e^2 e^{\left(\frac{\frac{-\epsilon^2}{\sum_{i=1}^{b} \sum_{j=1}^{i^\beta} 1}}{(\sum_{i=1}^{b} i^\beta)^2}\right)} = 2e^2 e^{-\epsilon^2 \sum_{i=1}^{b} i^\beta}.$$

Now, it can be clearly seen that $\|\theta_p^* - \hat{\theta}(b)\|_2 = \beta_\sigma \|X_{b,b^\beta}\|_2$ and let $\eta = \beta_\sigma \epsilon$. Then, $\forall \eta > 0$, we get

$$\Pr[\|\theta_p^* - \hat{\theta}(b)\|_2 \geq \eta] \leq 2e^2 e^{\frac{-(\sum_{i=1}^{b^\beta} i^\beta)\eta^2}{\beta_\sigma^2}}.$$

Using the Lipschitz property of expected reward function (Assumption 3), we have

$$\Pr(\exists\, x \in \mathcal{X} : |\bar{r}(x,\hat{\theta}(b)) - \bar{r}(x,\theta_p^*)| \geq \eta) \leq \Pr(R \cdot \|\theta_p^* - \hat{\theta}(b)\|_2 \geq \eta)$$

$$\leq 2e^2 e^{\frac{-(\sum_{i=1}^{b^\beta} i^\beta)\eta^2}{R^2\beta_\sigma^2}} \tag{12}$$

Taking complement of the event completes the proof. $\qquad\square$

## 7.1 Proof of Results in Section 3

### 7.1.1 Proof of Theorem 1

We first restate the theorem.

**Distribution Independent Regret:** When Algorithm PEGE is initialized with the parameters $C(a) = \log a$, $\alpha = 1/2$ and $\beta = 0$, and the online game is played over $T$ rounds, we get the following bound on expected regret:

$$R(T) \leq R_{max}|\sigma|T^{2/3} + 2R\beta_\sigma T^{2/3}\sqrt{\log 2e^2 + 2\log T} + R_{max} \tag{13}$$

where $\beta_\sigma$ is the constant as defined in Eq. 3.

*Proof.* Let Algorithm PEGE run for $K$ phases, with parameters initialized as $C(a) = \log a$, $\alpha = 1/2$ and $\beta = 0$.

**Exploration regret**: During every exploration phase, the expected regret is bounded by $|\sigma|R_{max}$, where $R_{max}$ is as given in Assumption 3. Thus, total expected regret due to exploration is $K|\sigma|R_{max}$.

**Exploitation regret**: Let $x^* \in \text{argmax}_{x \in \mathcal{X}}\, \bar{r}(x,\theta_p^*)$ and $x(b) \in \text{argmax}_{x \in \mathcal{X}}\, \bar{r}(x,\hat{\theta}(b))$. During every exploitation round within phase $b$ of Algorithm PEGE, the expected regret is $|\bar{r}(x(b),\theta_p^*) - \bar{r}(x^*,\theta_p^*)|$.

Now, from Lemma 5, with $\beta = 0$, the following holds w.p. $\geq 1 - \delta_b$,

$$\forall\, x,\ |\bar{r}(x,\theta_p^*) - \bar{r}(x,\hat{\theta}(b))| \leq \underbrace{\sqrt{\frac{R^2\beta_\sigma^2\log(\frac{2e^2}{\delta_b})}{b}}}_{\eta_b} \tag{14}$$

Then, w.p. $\geq 1 - \delta_b$, the following event holds true: $|\bar{r}(x^*,\theta_p^*) - \bar{r}(x(b),\theta_p^*)| \leq 2\eta_b$, as explained:

$$\bar{r}(x^*,\theta_p^*) \leq \bar{r}(x^*,\hat{\theta}(b)) + \eta_b\ \text{ from Eq. 14}$$
$$\leq \bar{r}(x(b),\hat{\theta}(b)) + \eta_b$$
$$\leq \bar{r}(x(b),\theta_p^*) + 2\eta_b\ \text{ from Eq. 14}$$

Thus, the event $|\bar{r}(x^*,\theta_p^*) - \bar{r}(x(b),\theta_p^*)| \leq 2\eta_b$ holds true w.p. $\geq 1 - \delta_b$, for every fixed phase $b$. Then, w.p. $\geq 1 - \sum_{i=1}^{K}\delta_i$, the following holds true:

$$\forall\, b,\ |\bar{r}(x^*,\theta_p^*) - \bar{r}(x(b),\theta_p^*)| \leq 2\eta_b$$

Note that the expected regret per round is always bounded by $R_{\max}$ (since expected reward is bounded by $R_{max}$).

The number of rounds of exploitation in phase $b$ is $b^\alpha$. Hence, the total expected regret due to exploitation, over $K$ phases is:

$$\sum_{i=1}^{K}\left(\underbrace{(1 - \sum_{j=1}^{K}\delta_j)\frac{2R\beta_\sigma\sqrt{\log(2e^2/\delta_i)}}{\sqrt{i}} + (\sum_{j=1}^{K}\delta_j)R_{\max}}_{\text{expected regret per exploitation round}}\right)i^\alpha$$

Taking $\delta_1 = \delta_2 = \ldots = \delta_K = \delta$, and summing over exploration and exploitation regret over $K$ phases, we get

$$R(T) \le K|\sigma|R_{max} + \sum_{i=1}^{K} \left( (1 - K\delta) \frac{2R\beta_\sigma \sqrt{\log(2e^2/\delta)}}{\sqrt{i}} + (K\delta)R_{\max} \right) i^\alpha \qquad (15)$$

Using the inequality $\sum_{i=1}^{K} i^y \le \int_0^K i^y dy \le K^{y+1}$, we get expected regret:

$$R(T) \le K|\sigma|R_{max} + (1 - K\delta)2R\beta_\sigma \sqrt{\log(2e^2/\delta)}K^{\alpha+1/2} + K\delta R_{\max}K^{\alpha+1} \qquad (16)$$

Now, we relate $K$ to total time $T$ as: $T = |\sigma|K + \sum_{i=1}^{K} i^\alpha \sim K^{\alpha+1}$, for large $K$.

Hence $K \sim T^{\frac{1}{1+\alpha}}$. Substituting value of $K$ in Eq 16, and taking $\alpha = 1/2$ and $\delta = \frac{1}{KT}$ gives us the required bound on expected regret. $\qquad \square$

Our next lemma shows that as the number of phases $b$ grows in Algorithm PEGE, the probability of selecting a sub-optimal arm for greedy exploitation shrinks.

**Lemma 6.** *At the end of exploration phase within phase $b$, $b = 1, 2, \ldots$, the estimator constructed is* $\hat{\theta}(b) = \frac{\sum_{i=1}^{b} \sum_{j=1}^{i^\beta} \tilde{\theta}_{j,i}}{\sum_{j=1}^{b} j^\beta}$. *Then the following holds,*

$$\Pr(\underset{x \in \mathcal{X}}{\operatorname{argmax}} \, \bar{r}(x, \hat{\theta}(b)) \not\subseteq \underset{x \in \mathcal{X}}{\operatorname{argmax}} \, \bar{r}(x, \theta_p^*)) \le 2e^2 e^{\frac{-(\sum_{i=1}^{b^\beta} i^\beta)\Delta^2}{4R^2\beta_\sigma^2}} \qquad (17)$$

*Proof.* Let us assume $x' \in \operatorname{argmax}_{x \in \mathcal{X}} \bar{r}(x, \hat{\theta}(b))$ such that $x' \notin \operatorname{argmax}_{x \in \mathcal{X}} \bar{r}(x, \theta_p^*)$. Let $x^* \in \operatorname{argmax}_{x \in \mathcal{X}} \bar{r}(x, \theta_p^*)$. Then, by our assumption, $\bar{r}(x', \hat{\theta}(b)) \ge \bar{r}(x^*, \hat{\theta}(b))$. By definition of gap $\Delta$, we also have $\bar{r}(x^*, \theta_p^*) - \bar{r}(x', \theta_p^*) \ge \Delta$. The two inequalities imply that at least one of the following two inequalities has to hold: either $|\bar{r}(x^*, \theta_p^*) - \bar{r}(x^*, \hat{\theta}(b))| \ge \frac{\Delta}{2}$ or $|\bar{r}(x', \hat{\theta}(b)) - \bar{r}(x', \theta_p^*)| \ge \frac{\Delta}{2}$.

Thus, $\operatorname{argmax}_{x \in \mathcal{X}} \bar{r}(x, \hat{\theta}(b)) \not\subseteq \operatorname{argmax}_{x \in \mathcal{X}} \bar{r}(x, \theta_p^*) \implies \exists x \in \mathcal{X} : |\bar{r}(x, \theta_p^*) - \bar{r}(x, \hat{\theta}(b))| \ge \frac{\Delta}{2}$. By using Lemma 5, and substituting $\eta = \frac{\Delta}{2}$, we get our result. $\qquad \square$

### 7.1.2 Proof of Theorem 2

We restate the theorem before proving:

**Distribution Dependent Regret:** When Algorithm PEGE is initialized with the parameters $C(a) = h \cdot a$, for a tuning parameter $h > 0$, $\alpha = 1$ and $\beta = 1$, and the online game is played over $T$ rounds, we get the following bound on expected regret:

$$R(T) \le \sum_{x \in \sigma} \Delta_x \left( \frac{\log T}{h} \right)^2 + \frac{4\sqrt{2\pi}e^2 R\Delta_{max}\beta_\sigma}{\Delta} e^{\frac{h^2(2R^2\beta_\sigma^2)}{\Delta^2}}. \qquad (18)$$

*Proof.* Let total number of phases that the algorithm runs for be $K$. We relate $K$ to total time $T$ as (after substituting parameters $C(a) = h \cdot a$, $\alpha = 1$ and $\beta = 1$ in Algorithm PEGE):

$$T = \sum_{i=1}^{K} |\sigma|i + \sum_{i=1}^{K} e^{hi} \ge e^{hK} \implies K \le \frac{\log T}{h}.$$

**Exploration regret**: Since we are in distribution dependent setting now, expected exploration regret in each exploration phase is $\sum_{x \in \sigma} \Delta_x$. Hence, total expected exploration regret is upper bounded by:

$$\sum_{i=1}^{K} (\sum_{x \in \sigma} \Delta_x)i = \sum_{x \in \sigma} \Delta_x \frac{K(K+1)}{2} \le (\sum_{x \in \sigma} \Delta_x) \frac{\log^2 T}{h^2}.$$

**Exploitation regret**: When a sub-optimal arm is picked in an exploitation round, the expected regret in that round is: $\leq \Delta_{max}$. Using Lemma 6 with $\beta = 1$, the total expected regret due to exploitation over $K$ phases is upper bounded by:

$$\sum_{i=1}^{K} \underbrace{2e^2\Delta_{max}\, e^{hi-\frac{i(i+1)}{2}\frac{\Delta^2}{4R^2\beta_\sigma^2}}}_{\text{expected exploitation regret upper bound in phase i}} \leq 2e^2\Delta_{max}\sum_{i=1}^{\infty} e^{hi-\frac{i(i+1)}{2}\frac{\Delta^2}{4R^2\beta_\sigma^2}}$$

$$\leq 2e^2\Delta_{max}\sum_{i=1}^{\infty} e^{hi-\frac{i^2}{2}\frac{\Delta^2}{4R^2\beta_\sigma^2}} \tag{19}$$

$$\leq 2e^2\Delta_{max}\int_{-\infty}^{\infty} e^{hy-\frac{y^2}{2}\frac{\Delta^2}{4R^2\beta_\sigma^2}}\, dy$$

The integral is the moment generating function (adjusting for normalization constant) of a gaussian random variable $Y \in \mathcal{N}(0, \frac{4R^2\beta_\sigma^2}{\Delta^2})$. Thus, the integral is $\mathbb{E}[e^{hY}] = e^{\frac{2h^2 R^2\beta_\sigma^2}{\Delta^2}}$ and total expected regret due to exploitation is upper bounded by: $\frac{4e^2\Delta_{max}\sqrt{2\pi}R\beta_\sigma}{\Delta}e^{\frac{2h^2 R^2\beta_\sigma^2}{\Delta^2}}$.

Summing over exploration and exploitation regrets completes the proof. $\qquad\square$

## 7.2 Proof of Results in Section 4

The following theorem is about the version of PEGE that Algorithm 3 calls on line 8. It will be needed in the proof of Theorem 4.

**Theorem 7. (Distribution Dependent Regret, version 2)** *When Algorithm 1 is initialized with the parameters $C(a) = h \cdot a$, for a tuning parameter $0 < h < \frac{\Delta^2}{4R^2\beta_\sigma^2}$, $\alpha = 1$ and $\beta = 0$, and the online game is played over $T$ rounds, we get the following bound on expected regret:*

$$R(T) \leq \sum_{x\in\sigma}\Delta_x\frac{\log T}{h} + \frac{2e^2\Delta_{max}}{\frac{\Delta^2}{4R^2\beta_\sigma^2} - h} \tag{20}$$

**Note**: Compared to Theorem 2, the regret bound has better dependence on $T$ — $O(\log T)$ instead of $O(\log^2 T)$ — but it also has a disadvantage. If the tuning parameter $h$ is incorrectly set, say $h \geq \frac{\Delta^2}{4R^2\beta_\sigma^2}$, then the bound does not even apply.

*Proof.* The proof is similar to proof of Theorem 2. We highlight the key steps:

Let total number of phases that the algorithm runs for be $K$. First: $T = \sum_{i=1}^{K}|\sigma| + \sum_{i=1}^{K}e^{hi} \geq e^{hK}$
$\implies K \leq \frac{\log T}{h}$.

Expected regret due to exploration: $\sum_{i=1}^{K}(\sum_{x\in\sigma}\Delta_x) = \sum_{x\in\sigma}\Delta_x K \leq (\sum_{x\in\sigma}\Delta_x)\frac{\log T}{h}$.

Expected regret due to exploitation: When a sub-optimal arm is picked, expected regret $\leq \Delta_{max}$. Using Lemma 6 with $\beta = 0$, and tuning parameter $h < \frac{\Delta^2}{4R^2\beta_\sigma^2}$, we get total expected regret due to exploitation

$$2e^2\Delta_{max}\sum_{i=1}^{K}e^{hi-i\frac{\Delta^2}{4R^2\beta_\sigma^2}} \leq 2e^2\Delta_{max}\sum_{i=1}^{\infty}e^{hi-i\frac{\Delta^2}{4R^2\beta_\sigma^2}}$$

$$= 2e^2\Delta_{max}\sum_{i=1}^{\infty}e^{-i(\frac{\Delta^2}{4R^2\beta_\sigma^2}-h)}$$

$$\leq 2e^2\Delta_{max}\int_{0}^{\infty}e^{-y(\frac{\Delta^2}{4R^2\beta_\sigma^2}-h)}dy \tag{21}$$

$$= \frac{2e^2\Delta_{max}}{\frac{\Delta^2}{4R^2\beta_\sigma^2} - h}$$

$\qquad\square$

### 7.2.1 Proof of Theorem 3

*Proof.* Note that Assumption 1 through Assumption 3 hold. Therefore, from Lemma 5, with $\beta = 0$ we get, with probability at least $1 - \delta_b$,

$$\forall x, \ |\bar{r}(x, \theta_p^*) - \bar{r}(x, \hat{\theta}(b))| \leq \sqrt{\frac{R^2 \beta_\sigma^2 \log(\frac{2e^2}{\delta_b})}{b}}$$

Let $\delta_b = \delta/2b^2$ which implies $\sum_{b \geq 1} \delta_b = \pi^2 \delta/12 < \delta$. Thus, setting $w(b) = \sqrt{\frac{R^2 \beta_\sigma^2 \log(\frac{4e^2 b^2}{\delta})}{b}}$, the event $E$ defined as

$$\forall b \geq 1, \forall x \in \mathcal{X}, |\bar{r}(x, \hat{\theta}(b)) - \bar{r}(x, \theta_p^*)| \leq w(b). \tag{22}$$

holds with probability at least $1 - \delta$.

1. Note that $b \geq T_1(\delta)$ implies $8w(b) < \Delta$. This is because the latter has the form $e^{Lb} > Mb$ with $M = 2e/\delta$ and $L = \Delta^2/(128R^2\beta_\sigma^2)$. Setting $b \geq 2/L \log(2M/L)$ guarantees that $e^{Lb/2} \geq 2M/L$ which implies that $e^{Lb} \geq Mb$ since $e^{Lb/2} \geq 1 + Lb/2 \geq Lb/2$.

   If $8w(b) < \Delta$ then clearly $2w(b) < \Delta$. Let $x \neq x^*$ be arbitrary. We have the following chain of implications:

   $$\begin{aligned}
   & 2w(b) < \Delta & \\
   \Rightarrow \quad & 2w(b) < \bar{r}(x^*, \theta_p^*) - \bar{r}(x_-^*, \theta_p^*) & \text{(def. of } \Delta) \\
   \Rightarrow \quad & 2w(b) < \bar{r}(x^*, \theta_p^*) - \bar{r}(x, \theta_p^*) & \text{(Assumption 4)} \\
   \Rightarrow \quad & 0 < \bar{r}(x^*, \hat{\theta}(b)) - \bar{r}(x, \hat{\theta}(b)). & (\because E \text{ holds})
   \end{aligned}$$

   This means that the If condition on line 12 will evaluate to true and $\hat{x}(b)$ on line 13 will be set to $x^*$.

   We also have the following chain of implications:

   $$\begin{aligned}
   & 8w(b) < \Delta & \\
   \Rightarrow \quad & 8w(b) < \bar{r}(x^*, \theta_p^*) - \bar{r}(x_-^*, \theta_p^*) & \text{(def. of } \Delta) \\
   \Rightarrow \quad & 8w(b) < \bar{r}(x^*, \theta_p^*) - \bar{r}(\hat{x}_-(b), \theta_p^*) & (\because \bar{r}(\hat{x}_-(b), \theta_p^*) \leq \bar{r}(x_-^*, \theta_p^*)) \\
   \Rightarrow \quad & 8w(b) < \bar{r}(\hat{x}(b), \theta_p^*) - \bar{r}(\hat{x}_-(b), \theta_p^*) & (\because \hat{x}(b) = x^*) \\
   \Rightarrow \quad & 6w(b) < \bar{r}(\hat{x}(b), \hat{\theta}(b)) - \bar{r}(\hat{x}_-(b), \hat{\theta}(b)). & (\because E \text{ holds})
   \end{aligned}$$

   This means that the If condition on line 15 will evaluate to true and the algorithm will stop and output an estimate $\hat{\Delta}$.

   Now suppose the algorithm stops and does not output "threshold exceeded" which means that the If conditions on line 12 and line 15 were both true at some episode $b$. Let $x \neq \hat{x}(b)$ be arbitrary. We have the following chain of implications:

   $$\begin{aligned}
   & 6w(b) < \bar{r}(\hat{x}(b), \hat{\theta}(b)) - \bar{r}(\hat{x}_-(b), \hat{\theta}(b)) & \text{(line 15)} \\
   \Rightarrow \quad & 6w(b) < \bar{r}(\hat{x}(b), \hat{\theta}(b)) - \bar{r}(x, \hat{\theta}(b)) & (\hat{x}(b) \text{ unique maximizer by line 12}) \\
   \Rightarrow \quad & 4w(b) < \bar{r}(\hat{x}(b), \theta_p^*) - \bar{r}(x, \theta_p^*). & (\because E \text{ holds})
   \end{aligned}$$

   This means, along with Assumption 4, that $\hat{x}(b) = x^*$. We also have,

   $$\begin{aligned}
   & 6w(b) < \bar{r}(\hat{x}(b), \hat{\theta}(b)) - \bar{r}(\hat{x}_-(b), \hat{\theta}(b)) & \text{(line 15)} \\
   \Rightarrow \quad & 6w(b) < \bar{r}(\hat{x}(b), \hat{\theta}(b)) - \bar{r}(x_-^*, \hat{\theta}(b)) & (\because \bar{r}(\hat{x}_-(b), \hat{\theta}(b)) \geq \bar{r}(x_-^*, \hat{\theta}(b))) \\
   \Rightarrow \quad & 4w(b) < \bar{r}(\hat{x}(b), \theta_p^*) - \bar{r}(x_-^*, \theta_p^*) & (\because E \text{ holds}) \\
   \Rightarrow \quad & 4w(b) < \bar{r}(x^*, \theta_p^*) - \bar{r}(x_-^*, \theta_p^*) & (\because \hat{x}(b) = x^*) \\
   \Rightarrow \quad & 4w(b) < \Delta. & \text{(def. of } \Delta)
   \end{aligned}$$

Now we prove that the output $\hat{\Delta}$ lies in the right range. We have

$$
\begin{aligned}
\hat{\Delta} &= \bar{r}(\hat{x}(b), \hat{\theta}(b)) - \bar{r}(\hat{x}_-(b), \hat{\theta}(b)) && \text{(line 16)} \\
&\geq \bar{r}(\hat{x}(b), \theta_p^*) - \bar{r}(\hat{x}_-(b), \theta_p^*) - 2w(b) && (\because E \text{ holds}) \\
&= \bar{r}(x^*, \theta_p^*) - \bar{r}(\hat{x}_-(b), \theta_p^*) - 2w(b) && (\because \hat{x}(b) = x^*) \\
&\geq \bar{r}(x^*, \theta_p^*) - \bar{r}(x_-^*, \theta_p^*) - 2w(b) && (\because \bar{r}(\hat{x}_-(b), \theta_p^*) \leq \bar{r}(x_-^*, \theta_p^*)) \\
&\geq \Delta - 2w(b) && \text{(def. of } \Delta) \\
&\geq \frac{\Delta}{2}. && (\because w(b) < \Delta/4)
\end{aligned}
$$

Similarly,

$$
\begin{aligned}
\hat{\Delta} &= \bar{r}(\hat{x}(b), \hat{\theta}(b)) - \bar{r}(\hat{x}_-(b), \hat{\theta}(b)) && \text{(line 16)} \\
&\leq \bar{r}(\hat{x}(b), \hat{\theta}(b)) - \bar{r}(x_-^*, \hat{\theta}(b)) && (\because \bar{r}(\hat{x}_-(b), \hat{\theta}(b)) \geq \bar{r}(x_-^*, \hat{\theta}(b))) \\
&= \bar{r}(x^*, \hat{\theta}(b)) - \bar{r}(x_-^*, \hat{\theta}(b)) && (\because \hat{x}(b) = x^*) \\
&\leq \bar{r}(x^*, \theta_p^*) - \bar{r}(x_-^*, \theta_p^*) + 2w(b) && (\because E \text{ holds}) \\
&\leq \Delta + 2w(b) && \text{(def. of } \Delta) \\
&\leq \frac{3\Delta}{2}. && (\because w(b) < \Delta/4)
\end{aligned}
$$

2. In this case $T_0 \leq T_1(\delta)$ but it could still be that the algorithm stops not because the threshold is exceeded but because line 12 and line 15 were true at some episode $b$. Clearly $b < T_0$, otherwise we would have output "threshold exceeded" and not produced an estimate $\hat{\Delta}$. Under the event $E$, the previous part shows that if stopping occurs with an estimate $\hat{\Delta}$, it must be that $4w(b) < \Delta$, i.e.

$$
4\sqrt{\frac{R^2 \beta_\sigma^2 \log(\frac{4e^2 b^2}{\delta})}{b}} < \Delta \quad \Rightarrow \quad b > \frac{16 R^2 \beta_\sigma^2}{\Delta^2} \log \frac{4e^2}{\delta} = T_2(\delta).
$$

This means $T_0 > b > T_2(\delta)$.

3. Finally, suppose Assumptions 1 through 3 hold but Assumption 4 fails. Event $E$ still holds with probability at least $1 - \delta$. However, if there are at least two optimal actions then, under $E$, their confidence intervals will always overlap and If condition on line 15 will never be true. That means that the algorithm can only stop when the threshold $T_0$ is exceeded.

$\square$

### 7.2.2 Proof of Theorem 4

*Proof.* We break the proof into the two cases mentioned in the theorem statement.

**Part 1: Assumption 4 fails.** From Theorem 3 we know, that with probability at least $1 - \delta$, Algorithm 2 outputs "threshold exceeded" in this case. Because of Eq. (22), we also have, for an optimal action $x^*$:

$$
|\bar{r}(\hat{x}(T_0), \theta_p^*) - \bar{r}(x^*, \theta_p^*)| \leq 2w(T_0)
$$

which implies a total regret of

$$
2w(T_0)(T - T_0|\sigma|) \leq 2w(T_0)T
$$

in the remaining $T - T_0|\sigma|$ rounds since we execute line 5. The regret when Algorithm 2 was running is bounded by $R_{max} T_0 |\sigma|$. On the bad event, which occurs with probability at most $1 - \delta$, the regret is at most $T R_{max}$ giving us a total expected regret of

$$
2w(T_0)T + T_0|\sigma|R_{max} + \delta T R_{max} = 2\sqrt{\frac{R^2 \beta_\sigma^2 \log(\frac{4e^2 T_0^2}{\delta})}{T_0}} T + T_0|\sigma|R_{max} + \delta T R_{max}
$$