[Reviews · NeurIPS 2016]

Reviewer 1

Summary

The authors deal with the stochastic combinatorial partial monitoring problem. In this setting, at each round an unknown vector theta is generated from an unknown fixed distribution, the operator chooses an arm among possibly infinitely many and receives a reward which is a known function of the arm and theta. Instead of receiving the reward as feedback or the full vector theta, the operator receives a known arm-dependent linear transformation of theta and tries to minimize its regret. The authors apply the Phased Exploration and Greedy Exploitation strategy to produce a first algorithm. This strategy consist ofÒ alternating between full exploration and greedy exploitation phases of predetermined length. They also present a version of this algorithm that first tries to estimate the gap. They then provide analysis for both algorithms and prove both problem-dependent and problem-independent regret bounds. These regret bounds enjoy the same complexity over T than previous work but are independent of the number of arms and do not require the assumption of a unique optimal arm in the problem-independent-bound case. They finally present an example of application to online ranking.

Qualitative Assessment

The paper is clear and well presented. The ideas to use the PEGE framework to completely get rid of dependency on the number of arms and to allocate samples to the gap estimation to get rid of the square in the problem dependent regret bound are interesting. I suggest to accept this paper. I have a few question for the authors. Q: You do not mention lower bounds for the regret. Do you know such bounds for the stochastic CPM problem and if so, how do they compare to your bound ? Q: You do not elaborate on the choice of the global observable set. Do you know a way to do a smart choice of the global observable set sigma so that \beta_\sigma is low ? AFTER I have read the rebuttal and kept my scores

Confidence in this Review

2-Confident (read it all; understood it all reasonably well)


Reviewer 2

Summary

The paper looks at the combinatorial version of partial monitoring games in stochastic settings. Two algorithms are developed based on forced exploration instead of confidence bounds. The first algorithm matches the state of the art distribution-free regret bound with a simpler oracle. The second matches the state of the art distribution-dependent regret bound. Both regret bounds depend on the size of a certain covering of the action set, not on its cardinality.

Qualitative Assessment

After rebuttal addendum: ------------------------------------ Rev_1: Line 249-253: The algorithm for “online ranking with top-1 feedback” was developed for the adversarial setting by the corresponding authors, specifically for the problem of ranking. What we show is that the problem setting can be subsumed in our general CPM setting, and thus, the specific “online ranking" problem can be solved even in stochastic setting, by the algorithm we have given for the general CPM problem. I find this response very confusing, especially the 'even' bit. So let me clarify: the stochastic adversary is less powerful than the oblivious adversary. I agree that you can deduce results for the stochastic instance of the ranking problem. But the existing algorithm was able to deal with oblivious adversaries too, and you are not as far as I can tell. Viewed in this light, this is a limitation of your current approach. I am still interested to know what possible additional benefit (there might be such) you get in return for making the stochastic iid assumption. This would be a matter of comparing bounds in terms of their dependency on the problem parameters. This was a minor comment that I am confident the authors can address. Original: ------------------------------------ This paper addresses an interesting and important problem on the current edge of online learning theory, and improves the dependence on certain relevant parameters (oracle requirement, dependence on action set). It discusses related work in detail, and points out how it relates to existing approaches. I would be happy to see this at NIPS. That said, please do have your manuscript proof-read for English grammatical and style issues. Especially focus on the use of articles 'the/a', plural, and where to place commas. Here are some minor remarks: line 117: S(x) should be S(\theta_p^*) line 118: \bar r is used before it is defined in Assumption 1, line 121. line 241. This would be an excellent place for commenting on the possibility of getting all the benefits of the various algorithms combined. What would be required to/currently prevents us from getting an anytime, argmax-oracle algorithm? lines 249-253: It seems that an algorithm for the non-stochastic (i.e. oblivious adversary) case is strictly more powerful/useful than a distribution independent stochastic algorithm. What is the advantage of your stochastic method? Do we get any improved dependencies on the problem parameters from your method? line 305: expectation misses \mathbb

Confidence in this Review

2-Confident (read it all; understood it all reasonably well)


Reviewer 3

Summary

The paper considers partial monitoring in the stochastic combinatorial setting with linear feedback mechanism, the version of the original problem that was introduced by Lin et al. in [1], and improves on some of the results in [1]. In particular, the paper proposes two algorithms, PEGE and PEGE2, for globally observable problems, both of which avoids some (but not all!) of the limitations of algorithm GCB of [1], including uniqueness constraint of the optimum, having access to oracles that solve (potentially) hard optimization tasks, and having a sample complexity that depends on the size of the action space.

Qualitative Assessment

The advantages of the PEGE algorithms, compared to GCB is that - they do not require an oracle that returns the second best solution of an optimization problem - they do not require the uniqueness of the optimal solution - their regret do not depend on the size of the action set. The disadvantages of the PEGE algorithms, compared to GCB are that - the claimed distribution dependent bound requires setting parameter h so that it satisfies some distribution-dependent conditions, which is not clear how to do efficiently if the parameters of the problem are not known - the distribution dependent bound has worse dependence on the time horizon T, and has an extremely bad (exponential) dependence on h. - to guarantee the claimed distribution dependent bounds requires different parameter setting than to guarantee the claimed distribution independent bounds. The issue with parameter h is a serious one, and thus the paper proposes a solution to it in form of another algorithm, PEGE2. However, the obtained algorithm does not provide a satisfactory solution to this issue, and suffers from other shortcomings too, as discussed below. Advantages, compared to GCB: - its regret does not depend on the size of the action set Disadvantages, compared to GCB: - the distribution dependent bound is not guaranteed in every case (see Equation (7)). It is not clear what kind of distribution dependent bound can be guaranteed in general. Finally, none of the proposed algorithms can avoid solving linear optimization problems in every step. In view of this, the results in the paper should be considered as interesting additions to our knowledge of the combinatorial partial monitoring problem, but it is not clear whether they take us any closer to an actual, effective solution of the problem. In particular, the obtained distribution dependent bounds seem particularly week. The strongest part, however, seems to be that the regret bounds do not depend on the size of the action set any more. It would be nice to highlight the main idea behind this in the main part of the paper. Additional remarks - The paper could benefit from adding a formal (algorithm-like) presentation of the protocol. - Please make it explicit that sigma in line 4 of Algorithm 1 is the globally observable set. - The remark in lines 176-180 is confusing. In what sense is the bound in [6] less explicit? Additionally, there should be an "is" between "in" and "not" in line 177. - Why do you introduce \delta, when it is always set to 1/T? - When adapting the combinatorial partial monitoring games to online ranking, how do the obtained results compare to existing methods? - "note -ve sign" in line 304 seems to be a typo. - Citation 4 in line 317 does not have all the authors names.

Confidence in this Review

2-Confident (read it all; understood it all reasonably well)


Reviewer 4

Summary

The paper first describes the existing model of combinatorial partial monitoring games in which the learner selects actions from an exponentially large set and the adversary samples moves from a bounded, continous space. Two algorithms are proposed to solve this problem taking inspiration from forced exploration techniques commonly used in the bandit literature. The first algorithm, called Phased Exploration with Greedy Exploitation (PEGE), does not require knowledge of the time horizon. The authors give both distribution dependent and independent upper bounds for PEGE. The second algorithm (PEGE2) tries to estimate the gap between the best and second best action to achieve better distribution dependent bound. The authors show that PEGE2 matches the upper bound of GCB, the state-of-the-art algorithm for CPM, removing the dependence on the size of the learner's action space. Finally, they motivate some application of the algorithmic framework to the problem of online ranking with feedback on the top.

Qualitative Assessment

The paper is very well written an easy to follow. The model of combinatorial partial monitoring games is clearly described. The main technical contributions of the paper are the PEGE and PEGE2 algorithms together with an analysis of their regret upper bounds. The authors consistently compare them to GCB as it is the only existing algorithm solving the exact same problem. The advantages of the proposed algorithms with respect to GCB are well motivated, as well as their drawbacks. However, I was disappointed no experiments was done to show the practical behaviour of PEGE/PEGE2 compared to that of GCB. The advantages of computing a single argmax are well motivated, but an experiment showing the regret behaviour of PEGE/PEGE2 with respect to GCB could have been insightful. At the end of the paper, the authors describe a potential application of their algorithms to a real-world problem, namely, the problem of online ranking with feedback at the top. The description of this application is welcome. The authors clame this limited feedback setting has pertinence in user privacy concern scenarii. This is not very clear to me. AFTER REBUTTAL I have read the rebuttal and kept my scores.

Confidence in this Review

1-Less confident (might not have understood significant parts)


Reviewer 5

Summary

The paper present two algorithms for a relatively new settings called combinatorial partial monitoring CPM. This setting is a combination of finite partial monitoring and combinatorial multi-armed bandit. The paper presents two algorithm against a stochastic adversary and provide theoretical regret bound (both the distribution dependent and independent one) for each of them. The first algorithm called PEGE combines the classical forced-exploration in MAB together with globally observable set to solve CPM. The second algorithm (PEGE2) first try to estimate the gap \Delta between the expected reward of the first best action (assumed unique) and that of the second best action. Then, this estimate is fed to PEGE. Finally, the paper discusses how this setting can be used in online ranking with feedback. The main improvement in the paper is to provide an algorithm that removes the assumption of the existence of a unique optimal action while still achieving the state-of-the art regret up to logarithmic factors.

Qualitative Assessment

The settings considered in the paper is nice and the provided algorithms are simple and well designed. I like the PEGE algorithm as it removed the non-reasonable assumption of a unique optimal action. 1- My only question is: Intuitively why would we get a problem dependent bound of O(log T) for PEGE2 if we set T_0 to O(T^{2/3})? 2- It will also be good to discuss this intuition in the paper

Confidence in this Review

2-Confident (read it all; understood it all reasonably well)


Reviewer 6

Summary

This paper proposed a new algorithm on partial monitoring games. Instead of using the UCB techniques in the previous CPM models, this paper use another classic technique "forcing with certainty equivalence". Their algorithm, PEGE, is based on a simpler oracle. This algorithm can also combine with gap estimation to achieve the same regret bound with previous result. The author also shows that this algorithm could be applied to online ranking.

Qualitative Assessment

This paper is 1. Some part of the paper is not clear. For example, abstract is too long and contains too many descriptions of the previous paper. 2. Since application(online ranking) is included, real world experiment result should be provided. Actually as sqrt(log T) is often much smaller than T^(2/3), the increase seems to be subtle. Maybe more justification should be added in this part?

Confidence in this Review

2-Confident (read it all; understood it all reasonably well)